# Barriers to conducting independent quantitative research in low-income countries: A cross-sectional study of public health graduate students in Liberia

**Hajah Kenneh**[1,2], **Tamba Fayiah**[3], **Bernice Dahn**[4], **Laura A. Skrip**[2,5]*

**1** Ministry of Health, Republic of Liberia, Monrovia, Liberia, **2** Quantitative-Data for Decision-Making Lab, Monrovia, Liberia, **3** PREVAIL, Monrovia, Liberia, **4** College of Health Sciences, University of Liberia, Monrovia, Liberia, **5** School of Public Health, College of Health Sciences, University of Liberia, Monrovia, Liberia

* skripla@ul.edu.lr

## Abstract

### Introduction

During recent disease outbreaks, quantitative research has been used to investigate intervention scenarios while accounting for local epidemiological, social, and clinical context. Despite the value of such work, few documented research efforts have been observed to originate from low-income countries. This study aimed to assess barriers that may be limiting the awareness and conduct of quantitative research among Liberian public health graduate students.

### Methods

A semi-structured questionnaire was administered September-November 2021 to Master's in Public Health (MPH) students in Liberia. Potential barriers around technology access, understanding of quantitative science, and availability of mentorship were interrogated. Associations between barriers and self-reported likelihood of conducting quantitative research within six months of the investigation period were evaluated using ordinal logistic regression.

### Results

Among 120 participating MPH students, 86% reported owning a personal computer, but 18.4% and 39.4% had machines with malfunctioning hardware and/or with battery power lasting ≤2 hours, respectively. On average, students reported having poor internet network 3.4 days weekly. 47% reported never using any computer software for analysis, and 46% reported no specific knowledge on statistical analysis. Students indicated spending a median 30 minutes per week reading scientific articles. Moreover, 50% had no access to quantitative research mentors. Despite barriers, 59% indicated they were very likely to undertake quantitative research in the next 6 months; only 7% indicated they were not at all

**Funding:** The author(s) received no specific funding for this work.

**Competing interests:** The authors have declared that no competing interests exist.

likely. Computer ownership was found to be statistically significantly associated with higher likelihood of conducting quantitative research in the multivariable analysis (aOR: 4.90,95% CI: 1.54–16.3).

## Conclusion

The high likelihood of conducting quantitative research among MPH students contrasts with limitations around computing capacity, awareness of research tools/methods, and access to mentorship. To promote rigorous analytical research in Liberia, there is a need for systematic measures to enhance capacity for diverse quantitative methods through efforts sensitive to the local research environment.

## Introduction

Research is essential for national development [1, 2]. Globally, the power, growth, and independence of nations are related to their capacity to produce and apply knowledge through research and innovation [3, 4]. That is because research generates new ideas and evidence to assist in curbing real-life and high-burden problems that stall progress [5].

In recent years, the need for evidence-based solutions to health threats has boosted the relevance and significance of research outputs. Immunologists and vaccinologists around the world have worked tirelessly to develop pharmaceutical interventions to protect against emerging and re-emerging pathogens such as Ebola virus and SARS-CoV-2 [6, 7]. In parallel, quantitative research methods, such as mathematical modeling, have provided evidence around intervention strategies utilizing pharmaceutical and non-pharmaceutical solutions [8, 9]. For instance, transmission models are being used to assess how SARS-CoV-2 transmission has varied over time and space [10–12], the likelihood of epidemics being seeded in new locations given introduction events [10], and coverage levels needed for interventions to ensure targeted reductions in morbidity and mortality such as to fall below existing intensive care capacity [13–15].

During public health emergencies, advances in technology, knowledge and skills have facilitated the rapid production of quantitative research output through increasingly complex methodologies—from big data analytics to bioinformatics to agent-based mathematical modeling [16, 17]. However, research and development associated with the adoption and utilization of such methods are not undertaken equally across settings [9, 15]. In particular, graduate students in Sub-Saharan African countries face challenges in conducting independent quantitative research [18, 19], and thus emerge from the educational pipeline without critical workforce competencies. As a result, the majority of research led by African students and professionals, including in Liberia, tends to be more qualitative and descriptive in nature or dependent on outside support for data analysis [18, 20, 21]. Limited, systematically collected data exist on this persistent problem and potential approaches to addressing it.

While short-term capacity-building efforts around research skills are frequently reported [22, 23], efforts to measure their impact are not often undertaken. Similarly important, there has been little formal investigation into core, context-specific barriers to the conduct of research, accounting for issues related to awareness of its utility as well as access to the tools required for undertaking it [20]. Elucidating such barriers would inform efforts to intervene more holistically and thus effectively for boosting research productivity in low-income settings.

In Liberia, the 2014–2015 Ebola outbreak brought attention to the need for in-country research capacity for local and global health security [24, 25]. Since then, investments have

sought to overcome challenges facing capacity-building in the post-conflict context (ex: [26]). Locally, the establishment of the National Public Health Institute of Liberia and of a Master of Public Health (MPH) program at the publicly funded University of Liberia have also provided dedicated environments for research and development as well as for the relevant training to undertake it. The MPH program is particularly focused on filling capacity gaps in data collection, management, and analysis despite infrastructural challenges and generally weak foundational understanding among students in quantitative sciences.

Here we describe the logistical, foundational, and mentorship-related barriers that impact public health graduate students' likelihood of undertaking high-quality and internationally competitive quantitative research in Liberia. This study thus aims to provide evidence around resource needs to enable tertiary- and graduate-level capacity for quantitative sciences in settings like Liberia with limited national resources for education and technology. Highlighting specific barriers to research capacity strengthening may inform targeted investments to provide an enabling environment for the development of critical competencies among students working or planning to work in the health sector, while the country undertakes longer-term, systemic improvements.

## Methods

### Study design and participants

A cross-sectional survey was conducted from September-November 2021 with public health graduate students from the two universities offering Masters of Public Health (MPH) degrees in Liberia. The selection of this population was due to their didactic and practical experiences with research as part of their MPH programs, and their direct experience with barriers that may be preventing them from undertaking research that requires rigorous quantitative methods.

### Data collection tool

The semi-structured survey consisted of mostly closed-ended questions with free-text fields for elaboration on barriers, potential solutions, and understanding of quantitative research methods. The majority of questions on the tool was based on the researchers' experience teaching and learning in Libera; approximately 20% of the questions on the tool were adapted from a survey previously used [21]. The survey consisted of two parts: part one included questions around the socio-demographic characteristics of the study participants, while part two interrogated potential barriers to the conduct of quantitative research. For the second part, questions about barriers were developed based on authors' experiences as public health students and faculty engaged in quantitative research in Liberia, as well as barriers identified in the literature [3]. Specifically, questions were developed to measure levels of foundational understanding of quantitative research methods, of access to resources needed for implementation of quantitative methods, and of support during the course of the research. In addition, the survey included a question about the likelihood of participants conducting independent quantitative research in the next six months. Low levels of understanding, access, and support were hypothesized to be barriers to the expressed likelihood of doing quantitative research.

To maximize data validity, pretesting of the survey was conducted with 20 medical students from the University of Liberia A.M. Dogliotti College of Medicine, which shares a campus with the University of Liberia School of Public Health (ULSOPH), and four students in the R Coding and Biostatistics Certificate Program co-administered by the ULSOPH and the Quantitative-Data for Decision-Making Lab. We documented on average how long it took the students to complete the tool and whether responses were consistent with expectation. Based on

observations during the pretest period, additional options were provided for multiple-choice questions, and open-ended questions leading to ambiguous or unstandardized responses were reworded or converted to closed-ended questions, respectively.

## Data collection procedure

At ULSOPH, mass communication about the survey was conducted via e-mail sent by the university administration to student groups. Additionally, the researchers visited core MPH classes, which are attended by entire cohorts as part of curricular requirements, to introduce the study and invite students to participate. For the Cuttington School of Graduate and Professional Studies, written communication was addressed to the Dean of the College of Health Sciences, where the MPH program is located, to introduce the researchers and the study and to request institutional participation. After permission was received, the researchers visited the campus twice per week (Friday and Saturday) for three consecutive weeks due to the high volume of MPH student attendance on those days.

All students had the option to complete the survey online or via paper forms. Initially, the survey was an electronic survey, disseminated as a Google form. However, a paper-based survey was made available after concerns were raised by potential participants expressing a lack of understanding of Google forms and/or limited internet access. After receipt of consent via electronic or paper form, follow-up emails or phone calls were sent to remind those who received the link to the survey but did not complete it or those who received a paper copy of the survey but did not return it.

## Sample size calculation

For the sample size calculation, it was assumed that the estimated proportion of students who would report being likely to conduct quantitative research even with challenges would be around 10%, compared with the proportion of students who would be likely to conduct quantitative research without challenges would be around 35%. Given a power of 80% and a significance level of 5%, it was determined that 120 participants would be required for the two-group comparison.

## Statistical analysis

Results were presented using descriptive statistics. Bivariable techniques, namely the chi-squared test or Fisher's exact test, were applied to assess factors/challenges associated with the likelihood of students engaging in quantitative research techniques. Univariable and multivariable ordinal analysis was also conducted to evaluate associations between different barriers and reported likelihood of conducting quantitative research in the next three months. The results generated were presented as ORs and 95% CI. The reference levels for categorical variables were the levels hypothesized to be associated with barriers and thus lower likelihood to engage in quantitative research. The likelihood of engaging in quantitative research was categorized as not at all likely, somehow likely, and very likely. The survey had allowed for four levels of likelihood (not at all likely, not very likely, a little likely, and very likely), but responses to the middle two options were collapsed into "somehow likely" for an ordinal logistic regression analysis. Univariable logistic regression investigated associations between the three-level likelihood variable and potential barriers. Barriers were found to be significantly associated with the likelihood of conducting quantitative research in the univariable analysis and were included in a multivariable model along with sex and age. The Brant test was used to assess the proportional odds assumption [27]. P-values $< 0.05$ were considered statistically significant. Analyses were conducted using R Statistical Software version 4.2.0.

### Ethics statement

Approval of the study protocol was obtained from the University of Liberia Institutional Review Board ahead of data collection. All participants provided written informed consent by signing their names to online or paper consent forms ahead of completing the survey.

## Results

### Overall

Overall, a total of 120 MPH students from two universities in Monrovia, Liberia, participated in the study, with 69.2% (83/120) completing the survey on paper and 30.8% (37/120) completing via an online Google form. Table 1 details characteristics of the survey participants. The highest proportions of students were aged 20–29 years (44/120, 36.7%) and 30–39 years (52/120, 43.3%). There was an approximately even distribution between male and female respondents, with slightly more males (62/120, 51.7%) than females (58/120, 48.3%). Out of the total respondents, 55.8% (67/120) attended the Cuttington School of Graduate and Professional Studies, while 44.2% (53/120) attended the University of Liberia School of Public Health. Applied Epidemiology was the area of public health emphasis for over 40% of the participating students (49/120, 40.8%).

### Computer and internet access

While most participants reported owning a personal computer (*i.e.*, laptop), 13.4% (16/119) did not (Table 2). Among those who own a personal laptop, about one-fifth (19/103, 18.4%) experienced hardware issues that warranted the use of an external keyboard or mouse.

**Table 1. Sociodemographic overview of the study sample.**

| Characteristic | N = 120[1] |
|---|---|
| **Age Group (Years)** | |
| 20–29 | 44.0 (36.7%) |
| 30–39 | 52.0 (43.3%) |
| 40–49 | 24.0 (20.0%) |
| **Gender** | |
| Female | 58.0 (48.3%) |
| Male | 62.0 (51.7%) |
| **Period of Admission** | |
| 1980–1989 | 1.0 (0.8%) |
| 1990–1999 | 2.0 (1.7%) |
| 2000–2009 | 29.0 (24.2%) |
| 2010–2019 | 88.0 (73.3%) |
| **Current University enrolled at** | |
| Cuttington University | 67.0 (55.8%) |
| University of Liberia | 53.0 (44.2%) |
| **Area of Concentration** | |
| Applied Epidemiology | 49.0 (40.8%) |
| Community Health | 17.0 (14.2%) |
| Environmental Health | 30.0 (25.0%) |
| Public Health Policy and Management | 24.0 (20.0%) |

[1]n (%)

**Table 2. Potential logistical barriers associated with the conduct of quantitative research methods.**

| Characteristic | n = 120[1] |
|---|---|
| **Ownership of personal computer** | 103 (86.6%)[2] |
| (No answer) | 1 |
| **Operating system on computer** | |
| Chromebook | 1 (1.0%) |
| IOS | 2 (2.1%) |
| Windows 7 | 11 (11.5%) |
| Windows 8 | 7 (7.3%) |
| Windows 10 | 74 (77.1%) |
| Windows 11 | 1 (1.0%) |
| (No answer) | 24 |
| **Duration of battery life per use** | |
| less than an hour | 9 (8.7%) |
| 1–2 hours | 29 (27.9%) |
| 3 hours | 32 (30.8%) |
| more than 3 hours | 34 (32.7%) |
| (No answer) | 16 |
| **Use of a computer that requires a different keyboard or mouse due to hardware issues** | 19 (18.4%) |
| (No answer) | 17 |
| **Amount of money (in USD) spent to access the internet (per day)** | $3.80 ± $3.10 |
| (No answer) | 5 |
| **On average, frequency of difficulty accessing the internet due to a poor network (days)** | 3.4 ± 2.1 |
| (No answer) | 3 |
| **On average, difficulty accessing the internet due to a poor network (days)** | 2.7 ± 2.0 |
| (No answer) | 2 |

[1]Variables are described either as n (%) for categorical variables or as mean ± standard deviation for continuous variables.

[2]Missing data for some variables may result in frequencies totaling less than the n = 120 sample size. % are based on n providing a response to the question.

Approximately 40% of participants had machines with batteries lasting two hours or less. 37% of participants (44/119) reported having issues with internet connectivity every day. On average, participants reported spending $3.80USD (SD: $3.10) per day on purchasing data to access the internet. Participants indicated that poor network affected their access to the internet an average 3.4 days per week (SD: 2.1), while lack of data (due to lack of funds) affected access an average 2.7 days (SD: 2.0).

## Understanding/Awareness of quantitative research and access to mentors

About half of participants indicated having specific knowledge about statistical analysis (60/111, 54.1%) or having used computer software to analyze data (62/115, 53.0%), although a higher percentage indicated that they had awareness of computer software used for data analysis in public health (73/115, 63.5%) (Table 3). Approximately 50% of participants reported access to a mentor or instructor who conducts quantitative research (58/117, 49.6%). Despite access to mentors and reported use of software for analysis, participants' understanding of quantitative research methods was limited. For instance, nearly two-thirds had no experience reading a scientific paper that used mathematical modeling as the methodology (74/116, 63.8%). Likewise, the time dedicated to learning or reading results from quantitative methods

**Table 3. Potential foundational and mentorship-related barriers associated with the conduct of quantitative research methods.**

| Characteristic | n = 120[1] |
|---|---|
| **Number of math or statistics classes taken in graduate school** | |
| 0 | 2 (1.7%)[2] |
| 1 | 23 (19.8%) |
| 2 | 39 (33.6%) |
| 3 | 13 (11.2%) |
| 4 | 26 (22.4%) |
| 5 or more | 13 (11.2%) |
| (No answer) | 4 |
| **No specific knowledge about statistical analysis** | 51 (45.9%) |
| (No answer) | 9 |
| **No awareness of any computer software used for data analysis in Public Health** | 42 (36.5%) |
| (No answer) | 5 |
| **No use of any computer software to analyze data** | 55 (47%) |
| (No answer) | 3 |
| **No experience reading a scientific paper on mathematical modeling** | 74 (63.8%) |
| (No answer) | 4 |
| **No access to a mentor or instructor who does quantitative research** | 59 (50.4%) |
| (No answer) | 3 |
| **Lack of experience in doing quantitative research** | 65.0 (56.5%) |
| (No answer) | 5 |
| **Low computer literacy** | 41.0 (35.7%) |
| (No answer) | 5 |

[1]n (%)

[2]Missing data for some variables may result in frequencies totaling less than the n = 120 sample size. %s are based on n providing a response to the question.

was relatively low for the study population. Respondents indicated spending a median 30 minutes per week (IQR: 0–120) reading scientific papers with statistical or other quantitative results.

Per the open-ended questions on the survey, many participants (almost uniformly) described quantitative research as "the process of collecting and analyzing numerical data." Only three participants indicated specific outcomes generated through quantitative methods (*e.g.*, "prevalence of disease, fertility rate of disease/condition, incidence rate of disease/condition"). When asked about methods, nearly all responses similarly referenced a survey or questionnaire (*e.g.*, "Descriptive survey method. . .that is gathered data from responses through questionnaires"). Two participants included mention of regression modeling techniques. Thus, it was observed that participants' understanding of quantitative research largely focused on the collection and description of data from closed-ended tools rather than a variety of analytical methods for utilizing primary or secondary data.

Nearly all participants (110/117, 94.0%) responded that they would be interested in training in the use of appropriate statistical tools of analysis, if the opportunity presented itself.

## Likelihood of conducting quantitative research

In general, there was a high reported likelihood of conducting quantitative research among MPH students in Liberia (Table 4). More than half of participants (69/115, 60.0%) indicated they were very likely to conduct quantitative research in the next six months, while 33.9% (39/

**Table 4. Associations between Likelihood of conducting quantitative research and challenges/barriers among MPH students in Liberia.**

| Characteristic | Overall, n = 120[1] | Not at all likely, n = 8[1] | Somehow likely, n = 41[1] | Very likely, n = 71[1] | p-value[2] |
|---|---|---|---|---|---|
| **No access to a mentor or instructor who does quantitative research** | 59 (50.4%) | 5 (71.4%) | 25 (61.0%) | 29 (42.0%) | 0.070 |
| **Lack of experience in doing quantitative research** | 65 (56.5%) | 6 (85.7%) | 18 (46.2%) | 41 (59.4%) | 0.107 |
| **Low computer literacy** | 41 (35.7%) | 6 (85.7%) | 10 (25.6%) | 25 (36.2%) | 0.009[3] |
| **Area of Concentration** | | | | | 0.015[3] |
| Applied Epidemiology | 49 (40.8%) | 0 (0.0%) | 17 (41.5%) | 32 (45.1%) | |
| Community Health | 17 (14.2%) | 2 (25.0%) | 4 (9.8%) | 11 (15.5%) | |
| Environmental Health | 30 (25.0%) | 6 (75.0%) | 9 (22.0%) | 15 (21.1%) | |
| Public Health Policy and Management | 24 (20.0%) | 0 (0.0%) | 11 (26.8%) | 13 (18.3%) | |
| **Number of math or statistics classes taken in graduate school** | | | | | 0.423 |
| 0 | 2 (1.7%) | 1 (12.5%) | 1 (2.5%) | 0 (0.0%) | |
| 1 | 23 (19.8%) | 3 (37.5%) | 6 (15.0%) | 14 (20.6%) | |
| 2 | 39 (33.6%) | 2 (25.0%) | 12 (30.0%) | 25 (36.8%) | |
| 3 | 13 (11.2%) | 1 (12.5%) | 4 (10.0%) | 8 (11.8%) | |
| 4 | 26 (22.4%) | 1 (12.5%) | 12 (30.0%) | 13 (19.1%) | |
| 5 or more | 13 (11.2%) | 0 (0.0%) | 5 (12.5%) | 8 (11.8%) | |
| **No ownership of personal computer** | 16 (13.4%) | 5 (62.5%) | 5 (12.2%) | 6 (8.6%) | 0.002[3] |
| **No specific knowledge about statistical analysis** | 51 (45.5%) | 5 (71.4%) | 22 (53.7%) | 24 (37.5%) | 0.190 |

[1]n (%)

[2]Calculated using the Fisher's exact test

[3]Statistically significant at p<0.05

115) reported they would be somehow likely and only 7/115 or 6.1% indicated that they would be not at all likely to conduct quantitative research. Statistically significant associations were identified between the likelihood of conducting quantitative research and the barriers of lack of computer skills and lack of computer ownership (Table 4).

In the univariable ordinal regression analysis, only two variables—computer ownership and access to mentors—were significantly associated with the outcome of reported likelihood of conducting quantitative research in the next six months (Table 5). For students who reported having computers for personal use, the odds of being more likely (*i.e.*, very likely or somehow likely versus not at all likely) to conduct quantitative research in the next six months was 4.6 times that of students who did not have computers for personal use (95% CI: 1.5–14.5, p = 0.01). Likewise, for students who reported having access to mentors, the odds of being more likely to conduct quantitative research in the next six months is 2.3 times that of students who do not have access to mentors (95% CI: 1.1–5.0, p = 0.03).

After adjusting for age, sex, and access to a mentor, the odds of being more likely (*i.e.*, very likely and somehow likely versus not at all likely) to conduct quantitative research in the next six months for students who reported having computers for personal use was 4.9 times that of students who reported that they do not have computers for personal use (95% CI: 1.54–16.3, p = 0.01) (Table 5). Current access to a mentor was not found to be significantly associated with likelihood of conducting quantitative research in the multivariable model.

## Discussion

Among public health graduate students in Liberia, there is a high likelihood of conducting quantitative research, yet the resources and foundational understanding to do so fully may be

**Table 5. Results of the ordinal regression model using three levels of likelihood of conducting quantitative research.**

| Characteristic | OR[1] | 95% CI[1] | aOR[1,2] | 95% CI[1] |
|---|---|---|---|---|
| **Age Group (Years)** | | | | |
| 20–29 | — | — | — | — |
| 30–39 | 1.15 | 0.52, 2.54 | 0.83 | 0.35, 1.94 |
| 40–49 | 2.19 | 0.79, 6.54 | 1.97 | 0.66, 6.41 |
| **Gender** | | | | |
| Female | — | — | — | — |
| Male | 1.17 | 0.57, 2.40 | 1.10 | 0.51, 2.34 |
| **Area of Concentration** | | | | |
| Applied Epidemiology | — | — | | |
| Community Health | 0.82 | 0.26, 2.71 | | |
| Environmental Health | 0.39 | 0.15, 0.99[3] | | |
| Public Health Policy and Management | 0.66 | 0.25, 1.75 | | |
| **Number of math or statistics classes taken in graduate school** | | | | |
| 0 | — | — | | |
| 1 | 17.1 | 1.06, 4.65[3] | | |
| 2 | 22.2 | 1.44, 5.84[3] | | |
| 3 | 19.3 | 1.10, 5.59[3] | | |
| 4 | 13.6 | 0.87, 3.59 | | |
| 5 or more | 21.7 | 1.24, 6.27[3] | | |
| **Lack of experience in doing quantitative research** | | | | |
| No | — | — | | |
| Yes | 1.16 | 0.55, 2.43 | | |
| **Low computer literacy** | | | | |
| No | — | — | | |
| Yes | 0.85 | 0.39, 1.87 | | |
| **Ownership of personal computer** | | | | |
| No | — | — | — | — |
| Yes | 4.61 | 1.53, 14.5[3] | 4.90 | 1.54, 16.3[3] |
| **Duration of battery life per use** | | | | |
| < 1 hour | — | — | | |
| 1–2 hours | 0.71 | 0.13, 3.13 | | |
| 3 hours | 0.95 | 0.17, 4.35 | | |
| > 3 hours | 0.74 | 0.13, 3.28 | | |
| **Amount of money (in USD) spent to access the internet (per day)** | 1.00 | 0.96, 1.05 | | |
| **On average, frequency of difficulty accessing the internet due to a poor network (days)** | 0.90 | 0.75, 1.07 | | |
| **Access to a mentor or instructor who does quantitative research** | | | | |
| No | — | — | — | — |
| Yes | 2.32 | 1.11, 4.95[3] | 2.03 | 0.94, 4.46 |

[1]OR = Unadjusted Odds Ratio, aOR = Adjusted Odds Ratio, CI = Confidence Interval

[2]Variables with all levels significant in the unadjusted analysis were included in the adjusted analysis.

[3]Statistically significant at $p < 0.05$

lacking. There is likewise great demand for training in advanced statistical methods. Compared to higher resource settings with dedicated programs of study for quantitative sciences, with high computer access and literacy among students, and with career pathways for quantitative researchers, Liberia is lagging [28]. No PhD programs in public health (or any other fields of study) exist in Liberia, and graduate-level coursework in mathematics and statistics is

limited in content and number. As efforts are underway to build quantitative research capacity in the public health workforce, understanding the research environment in sub-Saharan Africa [19] and innovatively bypassing contextual barriers in order to conduct rigorous research with the potential to inform decision-making (*e.g.*, [29–31]) is imperative.

Our findings reflect several potential technology-related challenges facing MPH students, including non-universal computer ownership, malfunctioning or poorly functioning hardware, outdated operating systems, and limited internet accessibility for effective and efficient conduct of quantitative research. These challenges impair the ability to utilize technologies, such as online student versions of statistical software or online repositories for syncing data and code, and are compounded by the local context of socioeconomic depression and lacking infrastructure. Of note, MPH students reported spending, on average, over $3USD daily on the internet which would amount to about $60USD monthly (for weekdays alone); this is considerable in a setting with a median monthly salary of 72,000LRD or under $475USD [32]. The investment in data for the internet contrasted with the reports of frequently poor network. Furthermore, while most participants indicated ownership of a personal computer, many faced the need to compensate for malfunctioning hardware or experienced inefficient battery power, which has implications for productivity given the recent unreliability of electricity access due to rationing, damaged infrastructure, and power theft [33]. Even so, these challenges were not found to be statistically significantly associated with the reported likelihood of the conduct of quantitative research. This is likely due to the requirement of conducting independent thesis research ahead of graduation.

We found that, within our study sample of MPH students in Liberia, there is almost universal interest in advanced training in quantitative research. However, the survey also demonstrated that current understanding of quantitative research is narrow in scope and that, despite some participants indicating that they are reading papers with mathematical modeling or statistical results, there is limited recognition of the methodologies. Liberian MPH students are also not engaging with scientific literature (our findings suggest they spend a median 30 minutes of reading scientific articles weekly) as graduate students in other settings might be. With this finding, there is a need to find contextually appropriate media for enhancing awareness about quantitative research, not only different methodologies but also their differing relevance across policy questions and processes [34, 35]. Such awareness across stakeholders will be essential for motivating simultaneous shifts to meet the needs for higher data quality [36] and more utilization of quantitative evidence [37] in Liberia.

Calls for investments in quantitative research capacity building are increasingly more common [22, 38]. While there has been some support given by external partners to build technical research capacity across sub-Saharan Africa through short courses or modules, the sustainability of the application of skills taught during short training opportunities is only occasionally considered [39]. In addition, the standardized curricula of such training does not always account for variation in foundational knowledge and technology access, which may leave some students behind as others progress through the program. Longer-term, locally owned, and contextually relevant training approaches could build the capacity needed to change practice and to enhance the availability of cadres of mentors—an essential element to sustainability of capacity-building in research [40]. Sufficient, in-depth capacity should be built as a way of increasing the pool of experts, while career opportunities to use the expertise should be prioritized for sustaining individual-level motivation and system-level gains.

The data collection tool and results presented here offer insights into potential barriers encountered by MPH students, but the study is not without limitations. For instance, there is opportunity to build upon the present findings to identify challenges that may be generalizable across graduate students from different disciplines. Furthermore, our data are limited to the

barriers that the researchers considered *a priori*. More nuanced findings around specific challenges revealed here may be necessary for actionable recommendations. For instance, future assessments of computer ownership as a challenge or potential barrier should include the understanding on how people use them because having access to personal computers does not mean they are being used for research purposes. There may be other barriers and solutions that the study did not cover; future studies should be conducted so as to provide more potential solutions in addressing the issue being studied. Qualitative research approaches could be used to solicit ideas.

## Conclusion

Locally led health research is crucial for understanding and identifying contextually relevant solutions for health challenges such as COVID-19. The findings of this study revealed that there is high interest and intent to conduct quantitative research among MPH students in Liberia. However, in addition to infrastructural and technological challenges to computer-based research, lack of mentorship, limited computer skills, and lack of prior knowledge and experience in quantitative research are prevalent. Such constraints presently impair the quality of such research. There is scope for harnessing the interest in quantitative research, yet training and professional opportunities should recognize the contextual barriers so that skills can be taught with understanding of existing knowledge and skills and sustainably utilized with existing resources.

## Supporting information

**S1 Checklist.**
(DOCX)

**S1 File.**
(XLSX)

## Acknowledgments

The authors acknowledge the students who provided feedback during the pretesting of the survey tool and the administrators at both the University of Liberia and Cuttington School of Graduate Studies who offered critical insight on the study objectives and procedures.

## Author Contributions

**Conceptualization:** Hajah Kenneh, Laura A. Skrip.

**Data curation:** Hajah Kenneh.

**Formal analysis:** Hajah Kenneh, Tamba Fayiah, Laura A. Skrip.

**Investigation:** Hajah Kenneh.

**Methodology:** Hajah Kenneh, Laura A. Skrip.

**Project administration:** Laura A. Skrip.

**Supervision:** Laura A. Skrip.

**Validation:** Tamba Fayiah, Laura A. Skrip.

**Writing – original draft:** Hajah Kenneh, Laura A. Skrip.

**Writing – review & editing:** Tamba Fayiah, Bernice Dahn, Laura A. Skrip.

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
