## [Decision Letter · Decision Letter 0]

2 Dec 2022

PONE-D-22-27103Barriers to conducting independent quantitative research in low-income countries: a cross-sectional study of public health graduate students in LiberiaPLOS ONE

Dear Dr. Kenneh H,

Thank you for submitting your manuscript to PLOS ONE. After careful consideration, we feel that it has merit but does not fully meet PLOS ONE’s publication criteria as it currently stands. Therefore, we invite you to submit a revised version of the manuscript that addresses the points raised during the review process.

We look forward to receiving your revised manuscript.

Kind regards,

Mohan Kumar

Academic Editor

PLOS ONE

Journal Requirements:

a) Did participants provide their written or verbal informed consent to participate in this study?

Additional Editor Comments:

A very relevant and well written manuscript. I would ask the authors to work on reviewer comments and re-submit the manuscript.

Reviewers' comments:

Reviewer's Responses to Questions

**Comments to the Author**

1. Is the manuscript technically sound, and do the data support the conclusions?

Reviewer #1: Yes

Reviewer #2: Yes

2. Has the statistical analysis been performed appropriately and rigorously? 

Reviewer #1: Yes

Reviewer #2: Yes

3. Have the authors made all data underlying the findings in their manuscript fully available?

Reviewer #1: Yes

Reviewer #2: Yes

4. Is the manuscript presented in an intelligible fashion and written in standard English?

Reviewer #1: Yes

Reviewer #2: Yes

5. Review Comments to the Author

Reviewer #1: The authors have submitted a well-written manuscript including ethical aspects. Methods, results, discussion, and conclusion are detailed and concise. The manuscript is acceptable in the current form for publication.

Reviewer #2: This study is potentially interesting, but its relatively simple design (cross sectional) and small sample limit the interest of the study.

I think that a qualitative study (focus group or interview) would have provided more interesting information than a quantitative study.

Some comments:

abstract:

It is necessary to add in this results section some aOR with their 95% CI, even if they are not significant.

Introduction: This introduction section should be more developed so that the reader understands the context, the issue and the interest of this study.

In addition, the objective of the study should be clearly defined at the end of this section.

Methods:

how did you choose the questions for the questionnaire? did you take them from other existing studies? or did you develop them specifically for this study? they were tested on a small group of students before being sent to the participants? more information should be given to the reader for the replication of the study

tables:

in table 5 if you have the 95% CI you don't need the p value

6. PLOS authors have the option to publish the peer review history of their article (what does this mean?). If published, this will include your full peer review and any attached files.

Reviewer #1: **Yes: **Denny John

Reviewer #2: No

---

## [Author Response · Author response to Decision Letter 0]

29 Dec 2022

Reviewer #1: 

1. The authors have submitted a well-written manuscript including ethical aspects. Methods, results, discussion, and conclusion are detailed and concise. The manuscript is acceptable in the current form for publication.

Thank you for the thoughtful feedback. We are eager to share the work and indeed hope that the results guide stakeholders interested in building research capacity in Liberia and other similar contexts.

Reviewer #2:

1. This study is potentially interesting, but its relatively simple design (cross sectional) and small sample limit the interest of the study. I think that a qualitative study (focus group or interview) would have provided more interesting information than a quantitative study.

Thank you for this comment. We expect that our findings will form the basis of ongoing research into barriers so that research capacity-building efforts are more informed. Larger studies with mixed method designs would indeed strengthen the evidence provided in our baseline investigation. We have commented on the need for further research and the value of qualitative data to provide more nuanced findings in the Limitations section of the Discussion.

2. Abstract: It is necessary to add in this results section some aOR with their 95% CI, even if they are not significant.

We appreciate the suggestion and have updated the last sentence of the Results section in the Abstract to include the relevant aOR and 95% CI. The sentence now states: “Computer ownership was found to be statistically significantly associated with higher likelihood of conducting quantitative research in the multivariable analysis (aOR: 4.90, 95% CI: 1.54 - 16.3).”

3. Introduction: This introduction section should be more developed so that the reader understands the context, the issue and the interest of this study.

The feedback on the Introduction section is appreciated. We have added a paragraph to provide more information about the research context in Liberia, and the interest of the study to provide understanding about the context for those interested in building research capacity, and/or designing investments to do so, in Liberia. Wehave likewise updated the final paragraph to explain our objective more explicitly.

4. Introduction: In addition, the objective of the study should be clearly defined at the end of this section.

We have added more clarity on the objective of the study at the end of the Introduction Section, which now includes the following statement: “The study aims to provide evidence around resource needs to enable tertiary- and graduate-level capacity for quantitative sciences in settings like Liberia with limited national resources for education and technology.”

5. Methods: How did you choose the questions for the questionnaire? did you take them from other existing studies? or did you develop them specifically for this study? 

Thank you for the comment. The selection of the questions for the questionnaire was two-fold. Approximately 20% of the questions in the questionnaire were adapted and revised from an existing study (Peter AM, 2017) while 80% were developed by the researchers. Within the Data Collection Tool section of the Methods, we have added this sentence: “The majority of questions on the tool was based on the researchers’ experience teaching and learning in Libera; approximately 5-10 questions (20%) on the tool were adapted from a survey previously used [21].”

6. Methods: They were tested on a small group of students before being sent to the participants? More information should be given to the reader for the replication of the study

The questionnaire was pretested on a small group of students from the Medical School and R Beginner students at Quantitative Data for Decision-Making Lab (Q4D-Lab). The pretest was intended to demonstrate participants' understanding of the questionnaire and to provide a measure of the time required for completion. The text included the number of persons completing the pretest and the action taken based on our observations during the process. However, to provide further detail, we have now added the following: “We documented on average how long it took the students to complete the tool and whether responses were consistent with expectation,” in the Data Collection Tool section of the Methods.

7. Tables: in table 5 if you have the 95% CI you don't need the p value

 The point is well taken. P-values have been removed from the Table.

---

## [Editor Report · Decision Letter 1]

12 Jan 2023

Barriers to conducting independent quantitative research in low-income countries: a cross-sectional study of public health graduate students in Liberia

PONE-D-22-27103R1

Dear Dr. Kenneh H,

We’re pleased to inform you that your manuscript has been judged scientifically suitable for publication and will be formally accepted for publication once it meets all outstanding technical requirements.

Kind regards,

Mohan Kumar

Academic Editor

PLOS ONE

---

## [Editor Report · Acceptance letter]

23 Jan 2023

PONE-D-22-27103R1 

*Barriers to conducting independent quantitative research in low-income countries: a cross-sectional study of public health graduate students in Liberia*  

Dear Dr. Kenneh:

I'm pleased to inform you that your manuscript has been deemed suitable for publication in PLOS ONE. Congratulations! Your manuscript is now with our production department. 

Kind regards, 

on behalf of

Dr. Mohan Kumar 

Academic Editor

PLOS ONE